# Caloric Restriction Can Ameliorate Postoperative Cognitive Dysfunction by Upregulating the Expression of Sirt1, MeCP2 and BDNF in the Hippocampal CA1 Region of Aged C57BL/6 Mice

**DOI:** 10.3390/brainsci13030462

**Published:** 2023-03-08

**Authors:** Lan Wei, Qiang Tao, Minmin Yao, Zhimeng Zhao, Shengjin Ge

**Affiliations:** 1Department of Anesthesiology, Shanghai General Hospital, School of Medicine, Shanghai Jiao Tong University, Shanghai 200000, China; 2Department of Anesthesiology, The International Peace Maternity and Child Health Hospital, School of Medicine, Shanghai Jiao Tong University, Shanghai 200000, China; 3Department of Anesthesiology, Zhongshan Hospital, Fudan University, Shanghai 200000, China

**Keywords:** calorie restriction (CR), postoperative cognitive dysfunction (POCD), silent information regulator 1 (Sirt1), methyl CpG binding protein 2 (MeCP2), brain derived neurotrophic factor (BDNF)

## Abstract

This study aimed to investigate the impact of caloric restriction (CR) on cognitive function in aged C57BL/6 mice after surgery, as well as the underlying mechanisms. Forty 14-month-old male C57BL/6 mice were randomly assigned to the ad libitum (AL, n = 20) group and the CR (n = 20) group. After feeding for 12 weeks, they were subdivided into four groups: AL control (ALC, n = 10), AL with surgery (ALS, n = 10), CR control (CRC, n = 10), and CR with surgery (CRS, n = 10). The Morris Water Maze (MWM) test was used to assess learning and memory capacity. By using western blot and immunofluorescence, the expression of Sirt1, MeCP2, and BDNF in the hippocampus and hippocampal CA1 region was quantified. According to the behavioral test, the CRC and CRS groups had significantly better learning and memory abilities than the ALC and ALS groups, respectively. Sirt1, MeCP2, and BDNF expression in the hippocampus and CA1 region in the hippocampus of the ALC and CRC groups of mice were correlated with cognitive improvement. In conclusion, CR could enhance the postoperative cognitive function in aged mice, most likely by increasing the expression of Sirt1, MeCP2, and BDNF in the CA1 region of the hippocampus.

## 1. Introduction

Postoperative cognitive dysfunction (POCD) is a common neurocognitive complication after surgery in the elderly, and is characterized by functional impairments, such as memory, executive function, direction, emotion, and visual–spatial structural ability [1,2]. According to recent research, elderly people over 65 years were highly likely to have POCD; moreover, the incidence can reach up to 47.6% of patients at one week and 34.2% of patients at three months after surgery [3]. There are numerous risk factors for POCD, including aging, educational level, inappropriate diet, obesity, anesthesia, surgery, etc. [4,5,6,7,8]. The main risk factor for POCD among them is aging, which is a sign of the gradual decline of physiological functions [4]. Unfortunately, despite extensive research on POCD for more than a century [1], due to the complex pathophysiology and mechanisms, there is still no effective preventive or therapeutic approach.

Early in 1935, McCay et al. discovered that dietary changes could prolong the survival time of mammals and decrease the incidence of aging-related diseases, like Alzheimer’s disease (AD) [9,10]. Research has shown that CR has positive effects on reducing neurodegenerative diseases and brain aging, in addition to extending lifespan [11,12]. CR in older adults has also been verified to improve recognition memory, which is closely related to hippocampal function [13]. As is common knowledge, the hippocampus plays a critical role in learning and memory [14,15]. Age-related cognitive deficits may be improved by dietary factors that enhance hippocampal neurogenesis and brain plasticity [16]. The hippocampus CA1 region is important for memory and learning, and cognitive dysfunction is closely linked to the activity of the neurons in this region [17,18,19]. CR has received increasing attention for its role in the cognitive abilities of the elderly. CR can be achieved by reducing calorie intake over a specific time period while maintaining adequate levels of macro- and micronutrients, and the typical levels of CR in mice and rats range from 10 to 50% [20,21,22].

Sirt1, an NAD+-dependent deacetylase that is expressed in neurons of the hippocampus, is essential for a number of brain processes, including normal learning, memory, and synaptic plasticity in mice [23]. Mice’s spatial learning and memory were weakened by hippocampal Sirt1 knockdown, which also caused hippocampal atrophy [24]. By altering Sirt1 expression in the mouse hippocampus, CR can have neuroprotective effects [25]. MeCP2 is a transcriptional regulator that is highly abundant in the brain and can bind to methylated genomic DNA to regulate a range of physiological processes that are associated with adult synaptic plasticity and neuronal development [26]. Hippocampal MeCP2 knockdown has the opposite effects of overexpression, which could improve synaptic plasticity and cognitive function [27]. However, less is known about MeCP2’s function in neurodegenerative diseases, such as AD and POCD.

BDNF is highly expressed in the brain, and it stands out for its broad roles in brain homeostasis, health, and disease via its complex downstream signals [28,29]. Numerous studies have firmly found that BDNF has a critical effect on hippocampal long-term potentiation (LTP), a process that prolongs synaptic efficacy and is thought to be the foundation of learning and memory [29]. BDNF deficiencies have been linked to neurological diseases like Huntington’s disease, AD, and POCD. But according to research, intermittent fasting (IF) and targeting cognitive performance to increase BDNF are two potential methods for enhancing brain health [30]. As a result, altering BDNF expression might be a workable way to reduce POCD.

CR has not only been shown in multiple studies to improve learning and memory in mice, but it can also activate Sir2/Sirt1, involving a number of molecular links, including nicotinamide adenine dinucleotide, nicotinamide, biotin, and related metabolites [25,31]. Previous research has indicated that BDNF in humans may be modulated by diet composition, such as CR [32]. Furthermore, SIRT1-mediated deacetylation of MeCP2 contributes to BDNF expression [33]. There is currently no evidence that CR can regulate the expression of MeCP2 or improve POCD by regulating the expression of Sirt1, MeCP2, and BDNF in the hippocampus. Therefore, this study aims to settle the following three issues: (1) whether CR can improve the POCD in aged C57BL/6 mice after surgery; (2) whether the expression of hippocampal Sirt1, MeCP2, and BDNF proteins changes after CR intervention; and (3) whether the responsible hippocampal region is CA1.

## 2. Methods

### 2.1. Animals and CR Model

Forty 14-month-old C57BL/6 male mice, weighing 36–47 g, were ordered from Shanghai SLAC (Shanghai, China, License number: 2008001622124). All experimental procedures were approved by the Ethics Committee of Zhongshan Hospital, Fudan University (Ethics number: 2017-0001). All mice were housed in separate cages and exposed to a clean environment. After acclimating for 1 week, they were randomly assigned to receive a regular diet ad libitum (AL group, n = 20) or a CR diet (CR group, n = 20) using a random number table. Mice in the AL group were fed 20 g of food regularly at 8:00 a.m. every day, and the residual food was weighed after 24 h. According to the literature, the average daily calorie intake was calculated and used as a guide to create a low-calorie diet with a 40% calorie reduction [34]. After 12 weeks of feeding, the mice were subdivided based on surgery performance: ad libitum control (ALC, n = 10), ad libitum with surgery (ALS, n = 10), CR control (CRC, n = 10) and CR with surgery (CRS, n = 10). Body weight was measured every Monday. Shanghai Pu Lu Teng Company (Q/VGBD1-2014, 25 kg) provided the regular and low-calorie food. The compositions of normal chow and low-calorie chow were displayed in Table 1. Protein, fat, carbohydrate, and other content in normal vs. low-calorie foods were 22.1% vs. 36.8%, 5.28% vs. 8.8%, 52% vs. 20%, and 20.62% vs. 34.4%, respectively. In order to prevent adverse events, the blood volume sampled is typically less than 10% of the total weight because the circulating blood volume makes up 6% of the total weight in normal adult mice. Given that a mouse’s total blood volume decreases with age, the blood-sampling period in this case was prolonged while the volume was maintained. To prevent unneeded stress reactions, blood samples were taken every four weeks, and the blood glucose level was measured.

### 2.2. Tail Vein Blood Glucose Test

The mice were fixed to expose their tails. After disinfection, the tail was submerged for 3–5 min in warm (45 °C) water to ensure adequate filling and vasodilation of local vessels. Blood was taken from the dorsal caudal vein by making a transverse incision at the distal third of the tail. An amount of 0.1 mL of blood was collected and the incision compressed with gauze to stop the bleeding. Blood samples were centrifuged at 12,000 rpm and 4 °C in a refrigerated centrifuge. After 5 min, the supernatants were collected and preserved at −80 °C. According to the assay kit’s instructions (F006; Nanjing Jiancheng Bioengineering Institute, Nanjing, China), a serum glucose test was performed. The serum glucose concentration was calculated using the following formula: Glucose (mmol/l) = (Experimental OD − Blank OD) × calibrant concentration (5.55 mmol/l)/[(Calibrant OD − Blank OD) × Experimental protein concentration (mgprot/mL)].

### 2.3. POCD Model

The temperature of the mice in the surgery groups was maintained at 37 °C on an electric blanket (yuyan am-92). According to the literature [35], for 4 h, mice were continuously given 2.8% isoflurane and 33% oxygen with the assistance of a small animal anesthesia unit to induce anesthesia. The end-tidal carbon dioxide partial pressure (P_ET_CO_2_), minimal alveolar concentration (MAC) of the inhaled anesthetics, and oxygen saturation (SpO_2_) were closely monitored. Following anesthesia and the loss of the righting reflex, open reduction internal fixation for tibial fractures was carried out using the modeling technique described in Terrando N’s study [36]. In short, the left lower limb’s knee region was shaved. The skin was opened 0.5 cm below the knee joint, exposing the tibia. To alleviate early postoperative pain, 0.3 mL of 2% lidocaine was used to induce topical anesthesia. Following that, a 0.38-mm steel needle was inserted and fixed into the bone marrow cavity along the long axis of the tibia. The mid-tibia was immediately clamped with surgical forceps, and the incision was sutured. Local application of anesthetic mixed with 2.5% Lidocaine and 2.5% Prilocaine, smeared at 8-h intervals within 48 h after surgery, provided analgesia. To avoid the spatial crowding effect interfering with the subsequent behavioral tests, mice in the control groups were anesthetized by breathing 33% oxygen for 4 h in the housing cage. Furthermore, P_ET_CO_2_ and SpO_2_ were not monitored in these mice to avoid additional stress.

### 2.4. Morris Water Maze Test

Daily behaviors and activities were fully recovered in mice 24 h after surgery. The typical MWM experiment was used to test mice’s capacity for spatial learning and memory. There were two tests: the navigation test and the spatial probe test. In the navigation test (6 d), the water phase was maintained at a height of 1.5 cm above the hidden platform. The escape latency period of mice, defined as the time from entering water to finding the hidden platform, indicated their spatial learning ability. The pool was divided into four quadrants, and mice facing the wall of the pool were randomly released into the water in each quadrant. Training was performed four times per day, and the results were averaged. At the end of each training, the mice were dried with paper towels, and the temperature was maintained at 37 °C under a medical heating lamp. The swimming trajectory was automatically recorded using the DigBehv-MM camera system. The maximum time allowed to find the hidden platform was 60 s. Success was determined when the mouse climbed onto the platform and remained there for 3 s. The escape latency period was recorded. The mouse was then left on the platform for 20 s to memorize the surrounding markers, which included triangle, square, circle, and rectangle markers, respectively, at four orientations. Training was subsequently completed on another mouse. The training intervals between mice were consistent. If the mouse failed to locate the hidden platform within 60 s, the mouse was manually guided to find the platform and stay there for 20 s. The corresponding escape latency period was recorded as 60 s. On the 7th day, the hidden platform was removed to perform the spatial probe test. Mice were placed in the pool at a random quadrant, and the time to cross the platform, swimming speed, and time spent swimming in the target quadrant within 60 s were recorded.

### 2.5. Tissue Sampling

After the behavioral test, mice were intraperitoneally injected with 1.5% pentobarbital sodium (0.1 mL/20 g, Batch No.57-33-0, Beijing Propbs Biotechnology Co., Ltd., Beijing, China). For western blot analysis, each group randomly selected 5 mice to obtain fresh hippocampus tissue, which was immediately stored at −80 °C. The other mice were perfused with 50 mL of 0.9% sodium chloride through the left apex. When the lungs, intestines, and liver became white, the sodium chloride injection was changed to 4% paraformaldehyde in phosphate buffered saline (PBS; 0.1 M, pH 7.4; 20 mL; fast to slow) until the organs and limbs became hard and the tail became stiff. The brain was then harvested and immersed in a 4% paraformaldehyde solution for fixation for 8 h. Subsequently, the brain was exposed to 15% and 30% sucrose solutions, successively, until completely dehydrated and sinking. The OCT-embedded frozen tissue block was prepared and sectioned using a freezing microtome (Leica CM1950) at 12 μm thickness. The sections were allowed to stay at room temperature for more than 30 min before being restored to a −20 °C refrigerator. Immunofluorescence staining was subsequently performed.

### 2.6. Western Blot

After extracting the protein, its concentration was measured by the bicinchoninic acid protein (BCA) method (Beyotime, Shanghai, China). The protein in each sample was denatured by boiling with a loading buffer (1 μL buffer per 4 μL protein sample) at 100 °C for 5 min, then transferred to a Polyvinylidene Fluoride (PVDF) membrane, with 5% skim milk covered to block unspecific bindings, on a shaker at room temperature. After 2 h, the membrane was washed with Tris-buffered saline with Tween 20 (TBST). Primary rabbit anti-mouse antibodies targeting β-actin (1:1000; Cell Signaling Technology, Danvers, MA, USA), Sirt1 (1:1000; Cell Signaling Technology), BDNF (1:1000, Abcam, Cambridge County, UK), and MeCP2 (1:1000; Cell Signaling Technology) were added at 4 °C for overnight incubation. On the following day, after TBST washing, alkaline phosphatase (AP) -coupled goat anti-rabbit secondary antibodies (1:10,000; Haimingrui Biotech, Beijing, China) were added at room temperature, and incubation was completed on a shaker for 2 h. The TBST washing procedure was repeated. ECL substrate (Haimingrui Biotech, Beijing, China) was employed to develop protein bands. Protein expression was qualitatively analyzed with the β-actin as the internal reference to indicate the relative expression. The protein bands were analyzed using the Image-Pro Plus software to calculate optical density (OD).

### 2.7. Immunofluorescence

The frozen tissue sections were washed 3 times with 1 × PBS. After that, the tissue was blocked with 1 × PBS, 10% goat serum, and 0.3% Triton at 37 °C for 2 h. Subsequently, the tissue was incubated with primary antibodies (Sirt1, 1:400, rabbit anti-mouse, Cell Signaling Technology; BDNF, 1:750, rabbit anti-mouse, Abcam Plc.; MeCP2, 1:200, rabbit anti-mouse, Cell Signaling Technology) at 4 °C overnight. On the next day, the tissue slides were washed 3 times with 1 × PBS. Then the tissue was incubated with secondary antibodies (1:1000; goat anti-rabbit IgG H&L, Abcam Plc.) in PBS in the dark for 1 h. 1 × PBS washing was performed again. The cell nuclei were stained with 4′,6-diamidino-2-phenylindole (DAPI), followed by 1 × PBS washing. Quenched fluorogens were added, and the slides were sealed. Images were captured under a fluorescence microscope, and the quantification of immuno-positive cells was analyzed using the Image-Pro Plus software version 6.0.

### 2.8. Statistical Analysis

All data were processed using the SPSS 23.0 statistical software, and the measurement data were expressed as mean ± standard deviation ($\bar{x}\pm \mathrm{sd}$). The body weight and blood glucose data were analyzed by Student’s *t*-tests. The food intake and calorie consumption data were analyzed with the Man–Whitney U test. The data from the behavior tests, the western blot, and immunofluorescence assays were analyzed using one-way ANOVA, and the LSD test was used for those with significant differences. A value of *p* < 0.05 was considered to be a statistically significant difference.

## 3. Results

### 3.1. Changes in Food Intake, Body Weight, Calorie Consumption and Blood Glucose in Aged Mice

The daily food intake of the CR group was significantly less than that of the AL group (Figure 1a). Differences in body weight began to occur between the two groups in the second week (*p* = 0.009) and became significant in the third week (*p* < 0.001) (Figure 1b). Comparatively, the body weight of mice in the CR group was remarkably lower than that of mice in the AL group. Both groups showed relatively stable weight changes at different time points. During the 12-week intervention period, the average CR rate of the CR group was 37.3% compared with the AL group (Figure 1c). Regarding blood glucose, it was slightly lower in the CR group versus the AL group, which was not statistically significant (Figure 1d).

### 3.2. Learning and Memory Ability in Aged Mice

In the MWM test, the decline of learning ability was manifested by the extension of escape latency (Figure 2a). When comparing the ALC and ALS groups, the escape latency period on days 3, 5, and 6 was statistically shorter in the ALC group (*p* < 0.05). Additionally, the period on days 3 and 6 was significantly shorter in the CRC group than that in the CRS group (*p* < 0.05), and that on days 3, 4, and 5 was remarkably shorter in the CRS group than that in the ALS group (all *p* < 0.05). When comparing the ALC group with the CRC group, the latency period on days 2, 3, 4, and 6 was relatively prolonged, with statistically significant differences (*p* < 0.05).

On the last day, memory was injured after surgery, as indicated by a significantly shorter aim-quadrant swimming time and fewer times crossing the platform. The aim-quadrant swimming time of the ALS group was the shortest compared with that in the other groups (*p* < 0.05, Figure 2b). The times of crossing the platform in the ALC and CRC groups was longer than that in the ALS and CRS groups, respectively (*p* < 0.05, Figure 2c). In addition, the times in the ALC group were less than those in the CRC group (*p* < 0.05, Figure 2c). The swimming speed of the ALC and CRC groups was slightly slower than that of the ALS and CRS groups, respectively, but there was no statistically significant difference (Figure 2d), which suggested that postoperative motor was independent from spatial learning and memory. The typical swimming patterns of four groups in the last hidden platform trial can be seen in Figure 2e.

### 3.3. Expression of Related Proteins in the Hippocampus

The results of the western blot showed that the level of BDNF in the ALC group was significantly higher than that in the ALS group (*p* < 0.05, Figure 3b), but significantly lower than that in the CRC group (*p* < 0.05, Figure 3b). Compared with the CRS group, the expression of BDNF was significantly higher in the CRC group (*p* < 0.05, Figure 3b). The expression of Sirt1 in the ALS group was significantly lower compared with the ALC group and the CRS group (*p* < 0.05, Figure 3c). On the contrary, the Sirt1 expression was much higher in the CRC group compared with the CRS group and the ALC group (*p* < 0.05, Figure 3c). The MeCP2 level in the ALC group was significantly higher than that in the ALS group (*p* < 0.05, Figure 3d), as well as in the CRC group compared with the CRS group (*p* < 0.05, Figure 3d). The level of MeCP2 in the CRC and CRS groups was higher than that in the ALC and ALS groups, respectively (*p* < 0.05, Figure 3d).

### 3.4. Expression of Related Proteins in Hippocampal CA1 Region

Consistent with the results of the western blot assay, the immunofluorescence staining results indicated that the optical densities of Sirt1-positive cells in the ALC group and the CRS group were significantly decreased compared with the CRC group (*p* < 0.05, Figure 4c). The Sirt1 level in the ALS group was significantly lower than that in the CRS and ALC groups (*p* < 0.05, Figure 4c). The BDNF (Figure 4d) and MeCP2 (Figure 4e) levels in the ALC group were significantly higher than those in the ALS group (*p* < 0.05), just as they were expressed in the CRC group compared with the CRS group (*p* < 0.05). The level of MeCP2 (Figure 4e) and BDNF (Figure 4d) in the CRC was higher than that in the ALC group (*p* < 0.05). When the ALS group was compared with the CRS group, the expression of MeCP2 remarkably decreased in the ALS group (*p* < 0.05, Figure 4e), while the expression of BDNF was not statistically different between the two groups.

## 4. Discussion

With improving global health and a steadily increasing elderly population, the frequency of surgery for progressive elderly patients and patients with a high prevalence of complications is increasing. Of note, POCD will be common in this patient population. The complications of POCD have been associated with a longer length of stay, higher mortality, and longer-term cognitive decline [4]. Therefore, this raises the need for a preventive strategy to facilitate early interventions for POCD during the perioperative period. As the main non-genetic mechanism, CR not only extends lifespan but also exerts neuroprotective effects. In this study, we used CR and POCD models to investigate the effects of CR on aged mice.

In the results, the body weight of the CR group was significantly lower compared with the AL group. The level of blood glucose was maintained simultaneously within a normal range. These results indicated that proper CR could suppress weight gain in healthy elderly mice while avoiding needless impairments to metabolism. This is consistent with the study of Quintas et al. [37], who discovered that CR significantly affected body weight while showing no effect on blood glucose. The model of tibial fracture fixation of POCD was modified from Terrando N [36], and the inhalation concentration of isoflurane was referred to N. Cesarovic [35]. Many kinds of surgical procedures could induce POCD, but we chose the tibial open fracture. Because this type of operation is the most common POCD model, this may be related to the high incidence of POCD in clinical orthopedic surgery [38]. In addition, orthopedic surgery is also a very common type of surgery in elderly patients. In the results of the MWM test, the swimming speed of different groups had no statistical significance, indicating that the mice recovered motor function after surgery. Other studies also demonstrated that an operation has no effect on swimming speed and the locomotor activity of the mice [39,40].

Our POCD model was valid, as evidenced by the ALS group and CRS group mice having shorter aim-quadrant swimming and times of crossing the platform than the ALC group and CRC group mice, respectively. In the behavioral test, the performance of the CRC group was better than that of the ALC group, indicating that CR could improve the learning and memory abilities in normal older mice, in accordance with those of many other studies [41,42]. The difference between the CRC group and the CRS group indicated that CR ameliorated POCD. The hippocampus plays a critical role in learning and memory. Region-specific analyses indicated that CA1 was more susceptible to aging stress, exhibiting a greater number of altered genes relative to CA3 and the dentate gyrus (DG) after CR [43].

Sirt1 has been recognized as a longevity gene and has been verified to be related to aging and disorders associated with it. Quantitative analysis showed that there was a significant reduction of Sirt1 protein levels in the hippocampus with aging [37]. While the long-term high expression of Sirt1 could completely maintain the cognitive ability in mice and delay the loss of neuronal synapse and the incidence of dysfunction [44]. The neuroprotective effect of Sirt1 may be related to the deacetylation of a variety of substrates, including peroxisome proliferator-activated receptor (PPAR)-γ Coactivator-1α (PGC-1α) and nuclear factor-κ B (NF-κ B) [45,46]. In the western blot assay and immunofluorescence test results of our research, the distinctions between the CRC group and the ALC group mice showed that CR enhanced the expression of Sirt1 in the hippocampus CA1 region of normal aged mice; the differences between the CRS group and ALS group mice indicated that CR-ameliorated POCD may be associated with the increased expression of Sirt1 in the hippocampus CA1 region. Strong evidence has shown that CR not only activates Sirt1, but also increases its expression to improve brain health [31,47].

During postnatal development, MeCP2 is necessary to maintain proper brain function. Rett syndrome (RTT), a kind of neurodevelopmental disorder that manifests as a variety of cognitive abnormalities, can result from mutations of the Mecp2 gene. Long-term memory formation in the adult hippocampus depends on MeCP2, which also maintains the chromatin features of mature CA1 neurons and can preserve the genomic responsiveness to hippocampal-dependent learning [48]. Research found that it was also emerging as a regulator of neuronal synaptic plasticity, and it could regulate the expression of BDNF via the interactions with miR-212 and miR-62132 [49,50]. Such regulation was shown to play a vital part in the functioning of the central nervous system and be key to maintaining the homeostasis of synapses and neurons [51]. The hypothesis of our study was that CR increases the expression of Sirt1, and Sirt1-mediated deacetylation of MeCP2 contributes to BDNF expression in hippocampus tissue [33]. Consistent with these findings, in the present study, the expression of MeCP2 in the CRC group was significantly higher than that in the ALC group. Combined with the behavior test and the lowest expression of MeCP2 being in the ALS group, we conclude that POCD may be alleviated by the increased expression of MeCP2 in the CA1 area of the hippocampus. In a study of stress-induced depressive mice, the expression changes of MeCP2, Sirt1, and/or neurotrophic factors in the hippocampus were consistent [52].

Widely present in the central nervous system, BDNF supports healthy neurodevelopment and is intimately linked to synaptic plasticity in neurons. In an animal study, it was discovered that exogenous BDNF could remarkably recover neuronal synapse function, whereas BDNF knockout caused major impairments in synaptic transmission and defects in the hippocampus LTP [53]. Given its function in synaptic repair, BDNF is considered a strategy for repair in neurodegenerative diseases [54]. Our results indicated that the level of BDNF was statistically lower in the ALC group compared with the CRC group. Therefore, CR-relieved cognitive function might be related to a higher level of BDNF in the hippocampal neurons. Surprisingly, this difference could not be found between the CRS group and the ALS group. But there was an increasing trend of expression in the CRS group. The factor leading to this difference may be related to our small sample size. Huang’s experimental data showed that CR could effectively promote the levels of BDNF in the hippocampus [55], which was similar to our results.

By now, CR has been studied for a number of years, and there exists a variety of treatment strategies. The majority suggest that CR is beneficial for the human body, either in extending the lifespan or protecting neurological functions. All in all, this study indicated a new idea and provided a theoretical basis for CR in the prevention and treatment of POCD. Nevertheless, there are limitations of this study. First, postoperative pain and its management are closely related to the occurrence of POCD [56]. Whether the application of an anesthetic cream (2.5% lidocaine and 2.5% prilocaine) could produce neuroprotective effects and reduce the incidence of POCD requires further investigation. Second, cholesterol levels were not monitored during feeding. Many studies have suggested that higher circulating TC is associated with an increased risk of cognitive impairment, especially in female subjects [57]. Therefore, further research is needed to confirm this point. Third, the behavioral test was conducted 1–7 days after surgery. The neuroprotective effects of CR for a long time after surgery demand further study.

## 5. Conclusions

To conclude, our study identified that preoperative CR could not only decrease body weight with normal concentrations of blood glucose but also provide a protective effect on the learning and memory abilities in elderly mice after surgery. This beneficial action was partially regulated through increased expressions of SIRT1, MeCP2, and BDNF in the hippocampus. Further, the level of these three proteins in the CA1 region of the hippocampus may be responsible for that process.

## Figures and Tables

**Figure 1 brainsci-13-00462-f001:**
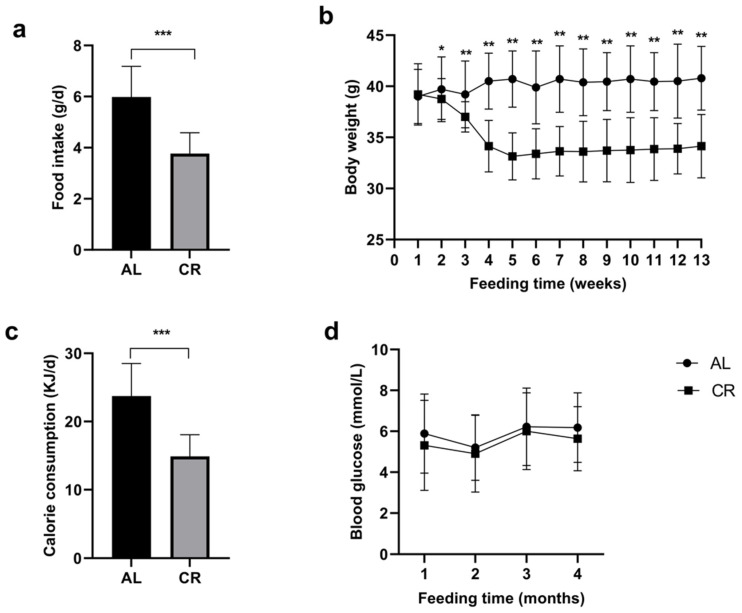
Food intake, body weight, calorie consumption, and blood glucose in aged mice of different groups; (**a**) Actual food consumption in different groups, determined every day; (**b**) The body weight in different groups, measured every week; (**c**) Calorie consumption in different groups, calculated based on food consumption; (**d**) Blood glucose level in different groups, measured every 4 weeks. * *p* < 0.05; ** *p* < 0.01; *** *p* < 0.001.

**Figure 2 brainsci-13-00462-f002:**
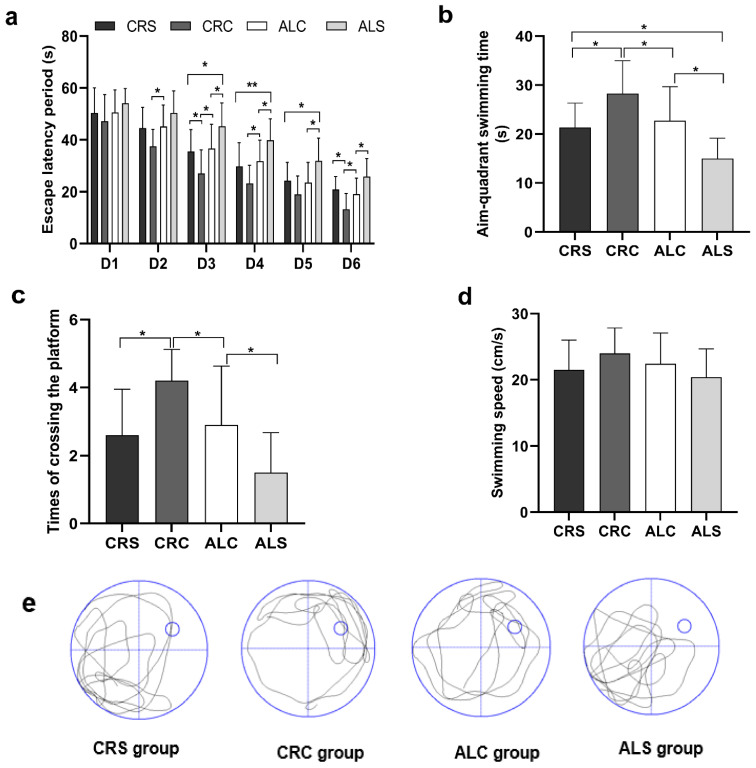
Learning and memory ability in aged mice of different group were assessed via the Morris Water Maze test. (**a**) Escape latency period obtained during Morris Water Maze navigation tests (from the 1st to the 6th day); (**b**) Aim-quadrant swimming time acquired during Morris Water Maze spatial probe tests (on the 7th day); (**c**) Times of crossing the platform gained on the 7th day; (**d**) Swimming speed derived on the 7th day; (**e**) Typical swimming patterns in the last hidden platform trial got on the 7th day. * *p* < 0.05; ** *p* < 0.01.

**Figure 3 brainsci-13-00462-f003:**
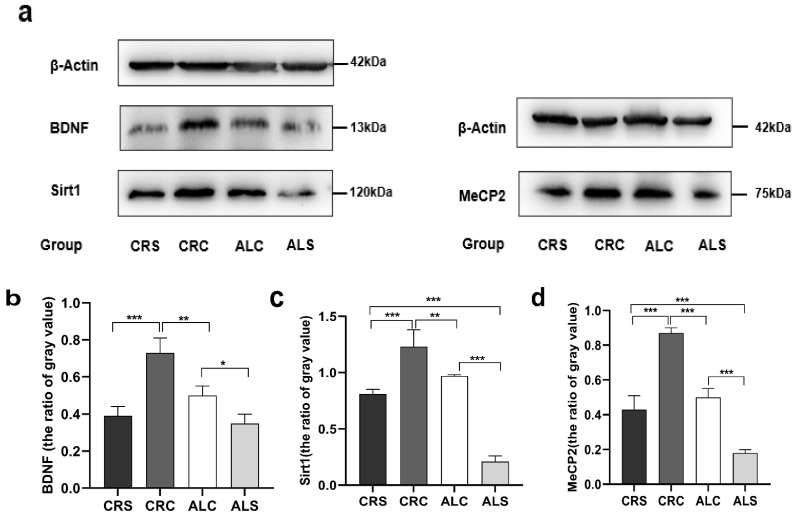
Expressions of Sirt1, BDNF, and MeCP2 in the hippocampus in aged mice of different groups. (**a**) Western blot bands indicate the expression of BDNF, Sirt1 MeCP2 and β-actin (as the internal control) in the hippocampus; (**b**–**d**) The corresponding quantitative results of Sirt1, BDNF, and MeCP2 shown at the bottom. * *p* < 0.05; ** *p* < 0.01; *** *p* < 0.001.

**Figure 4 brainsci-13-00462-f004:**
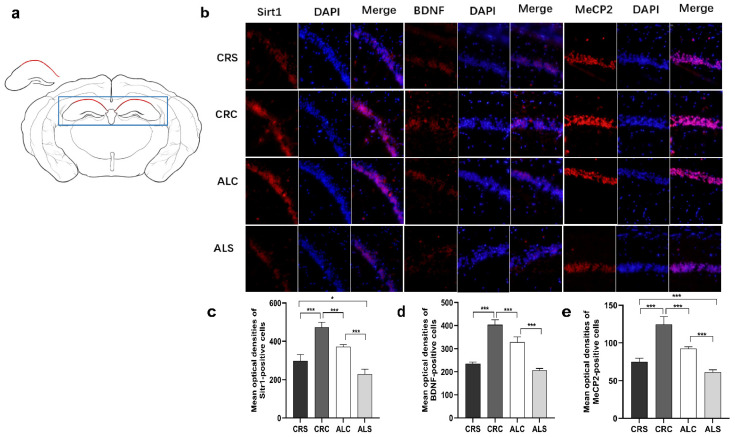
Expressions of Sirt1, BDNF, and MeCP2 in the hippocampal CA1 region in aged mice of different groups; (**a**) The drawing highlights the schematic diagram of hippocampal CA1 area, marked by the red line in the drawing; (**b**) The representative images for SIRT1, BDNF and MeCP2 in the hippocampal CA1 region; (**c**–**e**) The corresponding quantitative immune positive cells of Sirt1, BDNF, and MeCP2. Scale bar: 100 um. * *p* < 0.05; *** *p* < 0.001.

**Table 1 brainsci-13-00462-t001:** Composition of the diets.

Ingredient	Normal (g/100 g)	Low-Calorie Food (g/79.2 g)
Fish meal	4	4
Soybean meal	22	22
Corn starch	24	12.96
Wheat	34	18.36
Soybean	2	2
Soybean oil	2	2
Yeast	2	2
Bran	4	4
Grass flour	0.5	0.5
Premix	4.8	4.8
Choline	0.2	0.2
Maltodextrin	0.5	0.5
Total calories (KJ)	397	395

## Data Availability

Data can be made available by the corresponding author upon request.

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
