# Peer review of "Caloric Restriction Can Ameliorate Postoperative Cognitive Dysfunction by Upregulating the Expression of Sirt1, MeCP2 and BDNF in the Hippocampal CA1 Region of Aged C57BL/6 Mice"

_brainsci, 2023, doi:10.3390/brainsci13030462_

Round 1

Reviewer 1 Report

I congratulate the authors for conducting very interesting research. My comments are in the methods chapter to first present the design of the study, adopting the STOBE system. I believe the authors should add limitations and practical application of the study. I have doubts as to the correctness of the bibliography, please verify it in accordance with the guidelines of the journal

Author Response

Response to Reviewer 1 Comments

reviewer1.

Point 1: I congratulate the authors for conducting very interesting research. My comments are in the methods chapter to first present the design of the study, adopting the STOBE system. I believe the authors should add limitations and practical application of the study. I have doubts as to the correctness of the bibliography, please verify it in accordance with the guidelines of the journal.

Response 1:

We would like to thank you for giving us constructive suggestions which have help us in depth to improve the quality of the paper! We add limitations and practical application of the study in the last paragraph of the discussion. We checked the bibliography one by one and corrected it according to the guidelines of the journal, and replaced the incorrect one.

Reviewer 2 Report

The manuscript describes the importance of calorie restriction in treating post operative cognitive dysfunction through SIRT1, MeCP2 and BDNF in hippocampal CA1 region of aged C57BL/6 mice. Overall the manuscript is written well and the techniques handled were clear, with results neatly interpreted. However, there are few concerns

      Calorie restriction influencing the factors described in this manuscript has been well discussed in several research article over decades which actually diluted the present hypothesis. However, the post operative care through calorie restriction found logical.

      In the Introduction section, important roles of the target proteins such as BDNF and MeCp2 in the CNS can be highlighted to show the importance of BDNF and MeCp2 expression during post operative conditions.

      In spite of several surgical procedures, why internal tibial fracture fixation surgery was chosen to create the POCD model? Is there any other relevant POCD model available?

      The entire hippocampus is involved in spatial learning and memory. Specifically, CA1 was used to evaluate the expression of target proteins. What observations have been made in the CA2, CA3 and dentate gyrus regions?

      For immunofluorescence, better magnification of the images is required to showcase the results in an unambiguous manner and to arrive at a conclusion of the beneficial effects of calorie restriction.

      Furthermore, few typographical errors are seen in the results sections, which can be addressed by the authors.

Author Response

Response to Reviewer 2 Comments

Reviewer2

The manuscript describes the importance of calorie restriction in treating post operative cognitive dysfunction through SIRT1, MeCP2 and BDNF in hippocampal CA1 region of aged C57BL/6 mice. Overall the manuscript is written well and the techniques handled were clear, with results neatly interpreted. However, there are few concerns

Point 1:      Calorie restriction influencing the factors described in this manuscript has been well discussed in several research article over decades which actually diluted the present hypothesis. However, the post operative care through calorie restriction found logical.

Response 1:

We would like to thank you for giving us constructive suggestions which have help us in depth to improve the quality of the paper! The benefits of CR have been mostly observed in animal studies since 1935[1], in addition to extend lifespan, researches showed that CR has beneficial effects on alleviating neurodegenerative diseases and brain aging [2,3]. So, we propose the hypothesis that caloric restriction can improve postoperative cognitive impairment.

  1. McCay C M, Crowell M F, Maynard L A. The effect of retarded growth upon the length of life span and upon the ultimate body size[J]. JOURNAL OF NUTRITION, 1935, 10(1): 63-79.
  2. Zhang L, Xu H, Ding N, Li X, Chen X, Chen Z. Beneficial Effects on Brain Micro-Environment by Caloric Restriction in Alleviating Neurodegenerative Diseases and Brain Aging. Front Physiol. 2021 Nov 25;12:715443. [PubMed]
  3. Gillette-Guyonnet S, Vellas B. Caloric restriction and brain function. Curr Opin Clin Nutr Metab Care. 2008 Nov;11(6):686-92. [PubMed]

Point 2:     In the Introduction section, important roles of the target proteins such as BDNF and MeCp2 in the CNS can be highlighted to show the importance of BDNF and MeCp2 expression during post operative conditions.

Response 2:

Thank you! We rewrote the introduction. In the third and fourth paragraph of the introduction, we emphasized the important role of target proteins SIRT1, BDNF and MeCp2 in CNS to show the importance of SIRT1, BDNF and MeCp2 expression under postoperative conditions.

Point 3:      In spite of several surgical procedures, why internal tibial fracture fixation surgery was chosen to create the POCD model? Is there any other relevant POCD model available?

Response 3:

Thank you! In our study, the model of tibial fracture fixation of POCD was modified from Terrando N [1], and the inhalation concentration of isoflurane is referred to N. Cesarovic [2]. Many kinds of surgical procedures could induce POCD, but we chose the tibial open fracture because this type of operation is the most common POCD model, and this may be related to the high incidence of POCD in clinical orthopedic surgery [3]. In addition, orthopedic surgery is also a very common type of surgery in elderly patients.

1.Terrando N, Monaco C, Ma D, Foxwell BM, Feldmann M, Maze M. Tumor necrosis factor-alpha triggers a cytokine cascade yielding postoperative cognitive decline. Proc Natl Acad Sci U S A. 2010 Nov 23;107(47):20518-22.

2.N. Cesarovic, F. Nicholls, A. Rettichet al. Isoflurane and sevoflurane provide equally effective anaesthesia in laboratory mice. [J]. Laboratory Animals, 2010, 44(4): 329-336.

3.Wang CM, Chen WC, Zhang Y, Lin S, He HF. Update on the Mechanism and Treatment of Sevoflurane-Induced Postoperative Cognitive Dysfunction. Front Aging Neurosci. 2021 Jul 8;13:702231.

Point 4:      The entire hippocampus is involved in spatial learning and memory. Specifically, CA1 was used to evaluate the expression of target proteins. What observations have been made in the CA2, CA3 and dentate gyrus regions?

Response 4:

Thank you! Dietary factors can improve hippocampal neurogenesis and brain plasticity, which may improve age-related cognitive deficits [1]. In the hippocampus, the hippocampus CA1 region is vital for learning and memory, and the occurrence of cognitive dysfunction is closely related to the function of neurons in this area [2-4]. Region specific analyses indicated that CA1 was more susceptible to aging stress, exhibiting a greater number of altered genes relative to CA3 and the dentate gyrus (DG) after CR [5]. Therefore, we finally chose to evaluate CA1 area. Of course, it would be more scientific if other regions of the hippocampus were detected as control.

  1. Poulose SM, Miller MG, Scott T, Shukitt-Hale B. Nutritional Factors Affecting Adult Neurogenesis and Cognitive Function. Adv Nutr. 2017 Nov 15;8(6):804-811. [PubMed]
  2. Wang X, Li Y, Zhao J, Yu J, Zhang Q, Xu F, Zhang Y, Zhou Q, Yin C, Hou Z, Wang Q. Activation of astrocyte Gq pathway in hippocampal CA1 region attenuates anesthesia/surgery induced cognitive dysfunction in aged mice. Front Aging Neurosci. 2022 Nov 11;14:1040569. [PubMed]
  3. Ginsberg SD, Malek-Ahmadi MH, Alldred MJ, Che S, Elarova I, Chen Y, Jeanneteau F, Kranz TM, Chao MV, Counts SE, Mufson EJ. Selective decline of neurotrophin and neurotrophin receptor genes within CA1 pyramidal neurons and hippocampus proper: Correlation with cognitive performance and neuropathology in mild cognitive impairment and Alzheimer's disease. Hippocampus. 2019 May;29(5):422-439. [PubMed]
  4. Gu HF, Li N, Tang YL, Yan CQ, Shi Z, Yi SN, Zhou HL, Liao DF, OuYang XP. Nicotinate-curcumin ameliorates cognitive impairment in diabetic rats by rescuing autophagic flux in CA1 hippocampus. CNS Neurosci Ther. 2019 Apr;25(4):430-441.
  5. Zeier Z, Madorsky I, Xu Y, Ogle WO, Notterpek L, Foster TC. Gene expression in the hippocampus: regionally specific effects of aging and caloric restriction. Mech Ageing Dev. 2011 Jan-Feb;132(1-2):8-19.

Point 5:     For immunofluorescence, better magnification of the images is required to showcase the results in an unambiguous manner and to arrive at a conclusion of the beneficial effects of calorie restriction.

Response 5:

Thank you! For immunofluorescence, we enlarged the image and counted the mean optical densities of target protein-positive cells in figure 4.

Point 6:      Furthermore, few typographical errors are seen in the results sections, which can be by the authors.

Response 6:

Thank you! We have addressed the typographical errors in the results sections.

Reviewer 3 Report

The topic is interesting and the authors may have a working model for the effect of dietary interventions on perioperative outcome, although due to limitations in experimental design and, especially, in analysis and presentation, I am not convinced of that. To what extent animal models that can perform mobility-based tasks on the day following orthopedic surgery is a matter of discussion that should be addressed. It is quite striking that swimming speed does not differ between the groups. Is such immediate recovery of motor function typical for this model?

It is also unclear to me why the specific markers were used to correlate between behavior and molecular manifestations. No rational and no hypothesis to be tested by the biomarkers is clearly stated. 

In a future version, limitations of the experiments should be addressed separately.

However, there are unfortunately multiple substantial problems with all aspects of this MS in its current form. Below I only bring examples that are anything but complete.

1. Language: the issues range from minor (L.64) to major meaning the statements / intended claims are incomprehensible (L 89/90). These are just examples. The MS is riddled with these issues.

2. Analysis: it is not acceptable to present results as isolated p values for arbitrary comparisons. Results in Fig 2 are a good example. For multiple comparisons ANOVA or similar tests should be used not simply p values across multiple data columns.  Furthermore, please consider the following as an example: the control values e.g. CRC and ALC differ, therefore the question is whether CRS differs from CRC to a different degree than ALS from ALC. The presentation with numerous **** in the figure is not helpful. The figure legend is way too superficial and does not add much to understanding the results. The same must be said about the actual results. Meaningful comparisons are not presented in understandable, relevant form.

Similar critique applies to the other figures as well. Furthermore, to make any implications / correlation between behavior and biochemistry / cellular results and a specific brain structure is postulated, the claim must be supported by 'negative controls' e.g. if CA1 is critical for the interpretation of the link between behavior and biochem, then the same expression tests should be done on CA3 or another part of the brain to prove the specificity of the changes for CA1.

3. How much calories did the animals in the different groups consume daily or weekly?

Finally, the discussion is largely off-topic. Trying to cover the biological roles of SIRT1, BDNF, MeCP2 etc. with cherry-picked examples is completely unnecessary and distracting. Please stay on topic.

As an add-on: POCD is outdated terminology. There have been multiple consensus statements suggesting a more precise terminology.

Author Response

Author's Reply to the Review Report (Reviewer 3)

Reviewer3

The topic is interesting and the authors may have a working model for the effect of dietary interventions on perioperative outcome, although due to limitations in experimental design and, especially, in analysis and presentation, I am not convinced of that. To what extent animal models that can perform mobility-based tasks on the day following orthopedic surgery is a matter of discussion that should be addressed. It is quite striking that swimming speed does not differ between the groups. Is such immediate recovery of motor function typical for this model?

Response:

We would like to thank you for giving us constructive suggestions which have help us in depth to improve the quality of the paper! In this study, we established a POCD model, based on Terrando N’ [1] model which was a very mature model. We have also considered the problem of postoperative pain and postoperative movements, so we gave the treatment of postoperative pain. In addition, Ju H and Sun L also demonstrated that operation has no effect on swimming speed and the locomotor activity of the mice by the Morris water maze test and the Open-field test [36,37]. It was also observed that the locomotor activity, daily food and water consumption showed no abnormalities after 24 h after surgery in the ALS group and the CRS group in our study.

  1. Terrando N, Monaco C, Ma D, Foxwell BM, Feldmann M, Maze M. Tumor necrosis factor-alpha triggers a cytokine cascade yielding postoperative cognitive decline. Proc Natl Acad Sci U S A. 2010 Nov 23;107(47):20518-22.
  2. Ju H, Wang Y, Shi Q, Zhou Y, Ma R, Wu P, Fang H. Inhibition of connexin 43 hemichannels improves postoperative cognitive function in aged mice. Am J Transl Res. 2019 Apr 15;11(4):2280-2287. [PubMed]
  3. Sun L, Dong R, Xu X, Yang X, Peng M. Activation of cannabinoid receptor type 2 attenuates surgery-induced cognitive impairment in mice through anti-inflammatory activity. J Neuroinflammation. 2017 Jul 19;14(1):138.

It is also unclear to me why the specific markers were used to correlate between behavior and molecular manifestations. No rational and no hypothesis to be tested by the biomarkers is clearly stated. 

Response:

Thank you! We rewrote the introduction. In the third and fourth paragraph of the introduction, we emphasized the important role of target proteins SIRT1, BDNF and MeCp2 in CNS to illustrate why the specific markers were used to correlate between behavior and molecular manifestations.

In a future version, limitations of the experiments should be addressed separately.

Response:

Thank you! We add limitations and practical application of the study in the last paragraph of the discussion.

However, there are unfortunately multiple substantial problems with all aspects of this MS in its current form. Below I only bring examples that are anything but complete.

Point 1:   1. Language: the issues range from minor (L.64) to major meaning the statements / intended claims are incomprehensible (L 89/90). These are just examples. The MS is riddled with these issues.

Response 1:

Thank you! We have revised the introduction part and the discussion part to express the purpose and results of the study clearly. This research only preliminarily explored the relationship between CR and POCD, and the expression of related proteins in the hippocampus. Its internal mechanism needs further study.

Point 2:   2. Analysis: it is not acceptable to present results as isolated p values for arbitrary comparisons. Results in Fig 2 are a good example. For multiple comparisons ANOVA or similar tests should be used not simply p values across multiple data columns.  Furthermore, please consider the following as an example: the control values e.g. CRC and ALC differ, therefore the question is whether CRS differs from CRC to a different degree than ALS from ALC. The presentation with numerous **** in the figure is not helpful. The figure legend is way too superficial and does not add much to understanding the results. The same must be said about the actual results. Meaningful comparisons are not presented in understandable, relevant form.

Similar critique applies to the other figures as well. Furthermore, to make any implications / correlation between behavior and biochemistry / cellular results and a specific brain structure is postulated, the claim must be supported by 'negative controls' e.g. if CA1 is critical for the interpretation of the link between behavior and biochem, then the same expression tests should be done on CA3 or another part of the brain to prove the specificity of the changes for CA1.

Response 2:

Thank you! Firstly, the statistical comparison between groups in our experiment is not arbitrary but purposeful. The comparison of escape latency in different days is to illustrate the change trend of learning between groups, rather than simply list the statistically significant p values in each group. The difference between CRC and ALC is to explain the effect of CR on learning and memory of mice. If the degree of difference between CRS and CRC is different from that between ALS and ALC, it cannot be simply said that it is caused by the difference between CRC and ALC, because there is also the influence of surgical intervention. Moreover, the different degrees of CRS and CRC and the different degrees of ALS and ALC have no significance for this research and cannot explain any problem. ANOVA is applicable to the results of water Morris water maze, which is also the same in the literature [1]. Secondly, we have revised the figure legend from table1 to figure 4. Thirdly, it would be more scientific if other regions of the hippocampus were detected as control. However, we finally chose to evaluate CA1 area, because region specific analyses indicated that CA1 was more susceptible to aging stress, exhibiting a greater number of altered genes relative to CA3 and the dentate gyrus (DG) after CR [2].

  1. Tian H, Ding N, Guo M, Wang S, Wang Z, Liu H, Yang J, Li Y, Ren J, Jiang J, Li Z. Analysis of Learning and Memory Ability in an Alzheimer's Disease Mouse Model using the Morris Water Maze. J Vis Exp. 2019 Oct 29;(152). 
  2. Zeier Z, Madorsky I, Xu Y, Ogle WO, Notterpek L, Foster TC. Gene expression in the hippocampus: regionally specific effects of aging and caloric restriction. Mech Ageing Dev. 2011 Jan-Feb;132(1-2):8-19.

Point 3:   3. How much calories did the animals in the different groups consume daily or weekly?

 Response 3:

Thank you! We added figure1-c to show how many calories different groups of animals consume per day. calorie consumption in different groups were calculated based on food consumption.

Point 4:   Finally, the discussion is largely off-topic. Trying to cover the biological roles of SIRT1, BDNF, MeCP2 etc. with cherry-picked examples is completely unnecessary and distracting. Please stay on topic.

Response 4:

Thank you! We revised discussion which now covers the biological roles of SIRT1, BDNF, MeCP2.

Point 5:   As an add-on: POCD is outdated terminology. There have been multiple consensus statements suggesting a more precise terminology.

Response 5:

Thank you! The concept of PND was proposed in November 2018. PND includes all cognitive changes occurring before or within one year after surgery, including postoperative delirium (POD). In fact, most of the studies in basic research are the traditional POCD, that is the decline of learning and memory ability in a short time after surgery, and generally does not involve changes before surgery, nor changes as long as one year after surgery. Since there is no corresponding animal model, most basic experiments do not study postoperative delirium. Therefore, it is difficult to say clearly if PND was used in this research. From this perspective, POCD is more specific.

I searched PUBMED and found that 1,023 articles could be retrieved since 2018 if perioperative neurocognitive disorder (PND) was used as the keyword, while only 163 articles could be retrieved if animal studies were limited. Postoperative cognitive dysfunction (POCD) can be retrieved 2225 articles and 305 pieces if limiting to animal research. From this point of view, most scholars still agree with the concept of POCD.

Reviewer 4 Report

This article is generally interesting and my comments aim to increase the scientific soundness and clarity of it. However, some methodological issues must be clarified.

Line 77 – please write ad libitum in italics.

Line 102 – there are four tail veins: two lateral, one ventral and one dorsal. Which one was used?

Line 154 – Please provide details of pentobarbital sodium producer.

Line 201 – Did the authors test somehow the specificities of primary antisera used? Please describe. It looks that preadsorption experiments are missing.

Line 261 – Please submit full images of WBlots. Please change to “β-actin” not, “β-action”

Figure 4 – scale bars are missing.

Author Response

Author's Reply to the Review Report (Reviewer 4)

Reviewer4

This article is generally interesting and my comments aim to increase the scientific soundness and clarity of it. However, some methodological issues must be clarified.

Point 1: Line 77 – please write ad libitum in italics.

Response 1:

Thank you! We have altered ad libitum in italics.

Point 2: Line 102 – there are four tail veins: two lateral, one ventral and one dorsal. Which one was used?

Response 2:

Thank you! We used the dorsal one, and added it in 2.2. Tail vein blood glucose test.

Point 3: Line 154 – Please provide details of pentobarbital sodium producer.

Response 3:

Thank you! The details of pentobarbital sodium producer were provided in 2.5. Tissue sampling.

Point 4: Line 201 – Did the authors test somehow the specificities of primary antisera used? Please describe. It looks that preadsorption experiments are missing.

Response 4:

Thank you! In our study, the primary antibodies were selected by consulting the literature and the specificity was better in their results. Among them, the antibody of SIRT1 was applied basing on the research of Khan M [1] and slightly adjusted. The antibody of BDNF was used according to the experiment of Hu Y [2], and the experimental usage of Zhang L [3] is similar to ours. The application of MeCP2 is referred to Lee S and Choi J’s documents [4,5].

  1. Khan M, Ullah R, Rehman SU, Shah SA, Saeed K, Muhammad T, Park HY, Jo MH, Choe K, Rutten BPF, Kim MO. 17β-Estradiol Modulates SIRT1 and Halts Oxidative Stress-Mediated Cognitive Impairment in a Male Aging Mouse Model. Cells. 2019 Aug 19;8(8):928. 
  2. Hu Y, Zhang M, Chen Y, Yang Y, Zhang JJ. Postoperative intermittent fasting prevents hippocampal oxidative stress and memory deficits in a rat model of chronic cerebral hypoperfusion. Eur J Nutr. 2019 Feb;58(1):423-432.
  3. Zhang L, Song B, Zhang X, Jin M, An L, Han T, Liu F, Wang Z. Resveratrol Ameliorates Trigeminal Neuralgia-Induced Cognitive Deficits by Regulating Neural Ultrastructural Remodelling and the CREB/BDNF Pathway in Rats. Oxid Med Cell Longev. 2022 Nov 28;2022:4926678. 
  4. Lee S, Kim TK, Choi JE, Choi Y, You M, Ryu J, Chun YL, Ham S, Hyeon SJ, Ryu H, Kim HS, Im HI. Dysfunction of striatal MeCP2 is associated with cognitive decline in a mouse model of Alzheimer's disease. Theranostics. 2022 Jan 1;12(3):1404-1418. 
  5. Choi J, Kwon HJ, Lee JE, Lee Y, Seoh JY, Han PL. Hyperoxygenation revitalizes Alzheimer's disease pathology through the upregulation of neurotrophic factors. Aging Cell. 2019 Apr;18(2):e12888. 

Point 5: Line 261 – Please submit full images of WBlots. Please change to “β-actin” not, “β-action”

Response 5:

Thank you! These are our full images of WB blots. After the proteins were transferred to a Polyvinylidene Fluoride (PVDF) membrane. The target strip was cut and blocked by 5% skim milk at room temperature. Therefore, the bands are separated. We have changed “β-action” to “β-actin”.

Point 6: Figure 4 – scale bars are missing.

Response 6:

Thank you! Scale bars were added in the figure legend of figure 4.

Reviewer 5 Report

The manuscript could be improved by making some changes as suggested as follows;

In the abstract, it is mentioned as "aged mice" but did not mention the age or age range. It would be easier for the reader to clearly mentioned the age or age range.

The introduction section is well-written with the study background, aim etc. This study focused on three issues: 1) If CR could improve the POCD in aged C57BL/6 mice after surgery; 2) If the expression of hippocampal SIRT1, MeCP2, and BDNF protein changes after CR treatment; 3) If the responsible hippocampal region is CA1. And the authors really focused on these three issues.

Materials and methods are well described with all details.

Results:

Figure2A is very small and Crowdy, difficult to realize the significance status. Please improve it, and enlarge it.

In the case of western blot data, please mention the band size beside the blot images.

Figure4, please include an enlarged/magnified lane to show them clearly.

Author Response

Author's Reply to the Review Report (Reviewer 5)

Reviewer 5

Point 1: 

In the abstract, it is mentioned as "aged mice" but did not mention the age or age range. It would be easier for the reader to clearly mentioned the age or age range.

Response 1:

Thank you! We have changed "aged mice" into 14-month-old male C57BL/6 mice in the abstract.

Point 2: 

The introduction section is well-written with the study background, aim etc. This study focused on three issues: 1) If CR could improve the POCD in aged C57BL/6 mice after surgery; 2) If the expression of hippocampal SIRT1, MeCP2, and BDNF protein changes after CR treatment; 3) If the responsible hippocampal region is CA1. And the authors really focused on these three issues.

Materials and methods are well described with all details.

Results:

 Figure2A is very small and Crowdy, difficult to realize the significance status. Please improve it, and enlarge it.

Response 2:

Thank you! Figure2A has been enlarged and improved.

Point 3: In the case of western blot data, please mention the band size beside the blot images.

Response 3:

Thank you! The band size has been marked beside the blot images.

Point 4: Figure4, please include an enlarged/magnified lane to show them clearly.

Response 4:

Thank you! Figure4 has been magnified and improved.

Round 2

Reviewer 4 Report

The authors did not cooperate with a reviwever.

In the revised version none of my remark was implemented.

With regret I recommend the rejection of the manuscript.

Author Response

Dear Reviewer;

We feel great thanks for your professional review work on our article. As you are concerned, there are several problems that need to be addressed. According to your nice suggestions, we have made extensive corrections to our previous draft, the detailed corrections are listed below.

Point 1: Line 77 – please write ad libitum in italics.

Response 1:

We feel sorry for our carelessness. In our resubmitted manuscript, the type is revised. Thanks for your correction.

Point 2: Line 102 – there are four tail veins: two lateral, one ventral and one dorsal. Which one was used?

Response 2:

Thank you for your positive comments and valuable suggestions to improve the quality of our manuscript. We make a transverse incision at the distal third of the tail, and we used the dorsal one. This was added it in 2.2. Tail vein blood glucose test.

Point 3: Line 154 – Please provide details of pentobarbital sodium producer.

Response 3:

Thank you for your nice comments on our article. We have provided the details of pentobarbital sodium producer in 2.5. Tissue sampling (1.5% pentobarbital, Batch No.57-33-0,Beijing Propbs Biotechnology Co., Ltd).

Point 4: Line 201 – Did the authors test somehow the specificities of primary antisera used? Please describe. It looks that preadsorption experiments are missing.

Response 4:

We sincerely appreciate the valuable comments! In our study, the primary antibodies were selected by consulting the literature and the specificity was better in their results. Among them, the antibody of SIRT1 was applied basing on the research of Khan M [1] and slightly adjusted. The antibody of BDNF was used according to the experiment of Hu Y [2], and the experimental usage of Zhang L [3] is similar to ours. The application of MeCP2 is referred to Lee S and Choi J’s documents [4,5]. Therefore, we have not tested the specificities of primary antisera.

  1. Khan M, Ullah R, Rehman SU, Shah SA, Saeed K, Muhammad T, Park HY, Jo MH, Choe K, Rutten BPF, Kim MO. 17β-Estradiol Modulates SIRT1 and Halts Oxidative Stress-Mediated Cognitive Impairment in a Male Aging Mouse Model. Cells. 2019 Aug 19;8(8):928. 
  2. Hu Y, Zhang M, Chen Y, Yang Y, Zhang JJ. Postoperative intermittent fasting prevents hippocampal oxidative stress and memory deficits in a rat model of chronic cerebral hypoperfusion. Eur J Nutr. 2019 Feb;58(1):423-432.
  3. Zhang L, Song B, Zhang X, Jin M, An L, Han T, Liu F, Wang Z. Resveratrol Ameliorates Trigeminal Neuralgia-Induced Cognitive Deficits by Regulating Neural Ultrastructural Remodelling and the CREB/BDNF Pathway in Rats. Oxid Med Cell Longev. 2022 Nov 28;2022:4926678. 
  4. Lee S, Kim TK, Choi JE, Choi Y, You M, Ryu J, Chun YL, Ham S, Hyeon SJ, Ryu H, Kim HS, Im HI. Dysfunction of striatal MeCP2 is associated with cognitive decline in a mouse model of Alzheimer's disease. Theranostics. 2022 Jan 1;12(3):1404-1418. 
  5. Choi J, Kwon HJ, Lee JE, Lee Y, Seoh JY, Han PL. Hyperoxygenation revitalizes Alzheimer's disease pathology through the upregulation of neurotrophic factors. Aging Cell. 2019 Apr;18(2):e12888. 

Point 5: Line 261 – Please submit full images of WBlots. Please change to “β-actin” not, “β-action”

Response 5:

We feel great thanks for your professional review work on our article. We have submitted full images of Western Blots to the editor. In addition, “β-action ” is really a giant mistake to the whole quality of our article. We feel sorry for our carelessness. We have corrected it and we also feel great thanks for your point out

Point 6: Figure 4 – scale bars are missing.

Response 6:

Thanks for your careful checks. We are sorry for our carelessness. Based on your comments, we have added it in the figure legend of figure 4 (Scale bar: 100um).

We would like to take this opportunity to thank you again for all your time involved and this great opportunity for us to improve the manuscript. We hope you will find this revised version satisfactory.